# An Eye on *Staphylococcus aureus* Toxins: Roles in Ocular Damage and Inflammation

**DOI:** 10.3390/toxins11060356

**Published:** 2019-06-19

**Authors:** Roger Astley, Frederick C. Miller, Md Huzzatul Mursalin, Phillip S. Coburn, Michelle C. Callegan

**Affiliations:** 1Department of Ophthalmology, University of Oklahoma Health Sciences Center, Oklahoma City, OK 73104, USA; roger-astley@ouhsc.edu (R.A.); phillip-coburn@ouhsc.edu (P.S.C.); 2Department of Cell Biology and Department of Family and Preventive Medicine, University of Oklahoma Health Sciences Center, Oklahoma City, OK 73104, USA; frederick-miller@ouhsc.edu; 3Department of Microbiology and Immunology, University of Oklahoma Health Sciences Center, Oklahoma City, OK 73104, USA; MDHuzzatul-Mursalin@ouhsc.edu; 4Dean McGee Eye Institute, 608 Stanton L. Young Blvd., DMEI PA-418, Oklahoma City, OK 73104, USA

**Keywords:** toxins, enzymes, *Staphylococcus aureus*, eye, infection, in vivo models

## Abstract

*Staphylococcus aureus (S. aureus)* is a common pathogen of the eye, capable of infecting external tissues such as the tear duct, conjunctiva, and the cornea, as well the inner and more delicate anterior and posterior chambers. *S. aureus* produces numerous toxins and enzymes capable of causing profound damage to tissues and organs, as well as modulating the immune response to these infections. Unfortunately, in the context of ocular infections, this can mean blindness for the patient. The role of α-toxin in corneal infection (keratitis) and infection of the interior of the eye (endophthalmitis) has been well established by comparing virulence in animal models and α-toxin-deficient isogenic mutants with their wild-type parental strains. The importance of other toxins, such as β-toxin, γ-toxin, and Panton–Valentine leukocidin (PVL), have been analyzed to a lesser degree and their roles in eye infections are less clear. Other toxins such as the phenol-soluble modulins have yet to be examined in any animal models for their contributions to virulence in eye infections. This review discusses the state of current knowledge of the roles of *S. aureus* toxins in eye infections and the controversies existing as a result of the use of different infection models. The strengths and limitations of these ocular infection models are discussed, as well as the need for physiological relevance in the study of staphylococcal toxins in these models.

## 1. Introduction

Staphylococcal involvement in wound infections was established in the 1881 report by Ogston [1], who described the formation of new abscesses in guinea pigs and mice injected with pus taken from his patients. He called the clusters of bacteria he observed in the abscesses “staphyle” (“bunch of grapes” in Greek) because the arrangement of the bacterial cells resembled a cluster of grapes [2]. In 1884, Rosenbach [3] published studies on bacteria isolated from human wounds. He isolated and categorized two staphylococci by color: golden colonies were named *Staphylococcus aureus (S. aureus)*, and white colonies named *Staphylococcus albus* (now called *epidermidis*).

In 1928, *S. aureus* became infamous after the organism was identified as the cause of 18 children becoming violently ill after receiving diphtheria vaccinations in Bundaberg, Australia. Hours after inoculation, the children began to vomit, developed high fever, experienced convulsions, and fell into unconsciousness. Twelve of the children died within 36 h of vaccination. All of the surviving children developed severe abscesses at the injection site [4]. *S. aureus* was isolated from the vaccine vials [5]. The Commonwealth of Australia commissioned Burnet to study why the contamination had been so deadly [5]. Burnet demonstrated that supernatants from the *S. aureus* culture isolated from the vaccine caused hemolysis in vitro, caused skin-necrosis after intradermal inoculation, and was rapidly lethal to rabbits after intravenous injection. He concluded that a single heat-labile factor was responsible for these diverse pathological features [6]. Burnet further reported that other *S. aureus* isolates had the same pathological properties as the one isolated from the Bundaberg vaccine [7]. 

Burnet’s contention that the diverse disease features were due to a single toxin was disputed by various other researchers [8,9,10]. Glenny and Stevens [11] demonstrated that the hemolysis in vitro, skin-necrosis at the site of inoculation, and rapid lethality after intravenous injection were due to one immunologically distinct toxin, which they called “α-toxin”. Whether one toxin or more were responsible for these pathological features was not resolved until researchers were able to purify α-toxin [12,13] and demonstrate that α-toxin alone caused these pathological effects. Once purified toxin was available, researchers were able to study pore formation, identify α-toxin binding receptors, and examine how this single, potent toxin is able to produce such a diverse repertoire of actions.

*S. aureus* is a Gram-positive, nonmotile coccoid bacteria. *S. aureus* can be distinguished from other staphylococci by its classic gold pigmentation, β-hemolysis, a positive coagulase reaction, fermentation of mannitol, and a positive deoxyribonuclease test [14] (Figure 1). Although there are over 50 species of staphylococci, *S. aureus* is by far the most significant pathogen in this group [14]. *S. aureus* has an extensive array of virulence factors in its arsenal, which are depicted in Figure 2A. These virulence factors include proteins and enzymes associated with the bacterial surface which promote adherence to tissue and assist in biofilm formation, the cell wall itself which interacts with innate immune receptors to initiate acute inflammation, and a myriad of toxins and enzymes which promote bacterial spread by damaging tissue and lysing inflammatory cells. 

Although this collection of virulence factors renders *S. aureus* a formidable pathogen, this organism is part of the normal human flora. *S. aureus* nasal carriage among the general adult population varies depending on the geographical location [15]. In the Netherlands, the *S. aureus* carriage rate was reported to be 24–25.2% [16,17]. In the United States from 2001–2004, 30.4% of the population was reported to carry *S. aureus* [18]. Among healthy Japanese, 35.7% of individuals were carriers [19]. Saxena et al. [20] reported the carriage rate in India to be 29.4%. In Kenya, 18.3% of health care workers were *S. aureus* carriers [21]. In Mexico, 37.1% of healthy adults carried *S. aureus* [22]. Taken collectively, a quarter to one-third of individuals are carriers for *S. aureus* [23]. The most common site of colonization is the nasal mucosa, but *S. aureus* can also colonize the throat and the perineum [15,16,24,25].

Antibiotic resistance in *S. aureus* has emerged in waves, as *S. aureus* has counteracted the introduction of antibiotics as the primary treatment for bacterial diseases [26]. The first wave began after two patients were treated with penicillin in 1941 [27] and 1942 [27,28]. The first report of antibiotic resistance to penicillin was in 1944 [29], and by 1948, 59% of *S. aureus* cultures were reported to be resistant to penicillin [30], which is now know to be due to a plasmid-encoded penicillinase. *S. aureus* in this group were most commonly of phage type 80/81 [31]. By the 1950s, this phage group had become pandemic [26]. Phage type 80/81 seemingly disappeared after the introduction of methicillin as the preferred antibiotic for the treatment of staphylococcal infections [26,32].

The second wave of antibiotic resistance emerged with the introduction of methicillin in 1959 [33]. The first isolations of methicillin resistant *S. aureus* (MRSA) were reported in 1961 [34]. These *S. aureus* clones carried the *mecA* gene, which encodes the low-affinity penicillin binding protein PBP2a that imparts broad anti-beta-lactam resistance [35]. These isolates were mainly associated with hospitals and other health care establishments and are referred to as Healthcare-Associated MRSA (HA-MRSA). A third wave of antibiotic resistance occurred with the development of methicillin-resistant strains of *S. aureus* outside of health care establishments, called Community-Associated MRSA (CA-MRSA). These linages are unrelated to HA-MRSA and arose from numerous genetically distinct lineages across the world. Some have spread internationally while others have remained restricted to certain geographic regions [36,37,38]. In the United States, USA300 is the most prevalent strain of CA-MRSA, but each continent appears to have its own predominant type [26].

## 2. *S. aureus* and Eye Infections

*S. aureus* is a leading cause of eye infections such as dacryocystitis, conjunctivitis, keratitis, cellulitis, corneal ulcers, blebitis, and endophthalmitis [39,40,41]. Contact lens wearers, especially those who wear extended-wear contact lenses or fail at proper lens hygiene, are at greater risk for the development of keratitis. *S. aureus* is one of the leading causes of bacterial keratitis, along with *Pseudomonas aeruginosa.* The number of eye infections appears to be rising due to increasing use of ocular surgical procedures, especially in elderly and diabetic populations. In the developed world, the rate of intraocular surgeries, for example for cataract removal and lens replacement, has been steadily increasing [42]. The leading infectious complication from cataract surgery is endophthalmitis, with an incidence of 0.01–0.3% [43,44]. Intravitreal injections are also being performed at greater frequencies for the treatment of neovascular eye disease, neurodegenerative diseases, and intraocular inflammation [45]. It was estimated that in 2014, 18 million intraocular injections of anti-vascular endothelial growth factor (VEGF) agents to treat age-related macular degeneration were performed [46]. The frequencies of injection-related complications has also increased [47], with endophthalmitis incidences after intravitreal injection reported to be 0.006–0.16% per injection and 0.7–1.3% over the course of treatment [48,49,50,51]. While these infection rates are statistically very low compared to diseases such as diabetes or cancer, because of the numbers of procedures, they number in the tens of thousands. In addition, eye infections can cause profound vision loss and disability, costing millions of healthcare dollars and reductions in quality of life for those affected. 

*S. aureus* expresses an amalgam of toxins, enzymes, and secreted proteins whose virulence in humans and animal models has been well documented [13,52,53,54,55]. Studies on the contribution of these virulence factors to eye infections have focused on a few well-characterized toxins, such as α-toxin, β-toxin, γ-toxin, and PVL (Figure 2B). However, *S. aureus* has many more toxic factors in its armamentarium, some whose roles in eye infections have yet to be defined. Most studies of the activities of *S. aureus* toxins have focused on cell killing, but many recent studies have indicated that sublytic doses of some *S. aureus* toxins can have dramatic effects on target cells [56,57,58,59]. These reports have indicated that by altering the permeability of cell membranes, toxins can kill or manipulate the functions of the immune cells, and disrupt epithelial barriers to promote bacterial growth and spreading. By interacting with various cell surface proteins, toxins are able to target diverse cell types—mostly cells of the immune system which only express those receptors [56,57,58,59,60]. 

*S. aureus* elaborates many virulence factors which, depending upon the infection site in the eye, can manifest as serious but treatable infections such as conjunctivitis or dacryocystitis, or sight-threating infections such as corneal ulcers, endophthalmitis, or orbital cellulitis [39,40,41]. The eye is an especially vulnerable organ whose integrity, structure, and function are essential for proper sight. For the most part, our eyes resist the negative impact of infecting organisms through the concerted actions of blinking, tear film, and antimicrobial peptides and other enzymes which protect the ocular surface [55,61,62,63]. When *S. aureus* comes in contact with tissues of the eye and is able to circumvent these protections, infection results (Figure 2B). This review will discuss *S. aureus* and its toxins in the context of ocular infections and examine studies conducted in animal models to determine the roles and mechanisms of *S. aureus* toxins in eye infections. The strengths and limitations of these models and future steps for study will be discussed.

## 3. Membrane-Damaging Toxins

α-toxin is also known as α-hemolysin because it was originally named for its ability to cause β-hemolysis of red blood cells [11]. Generations of microbiology students are therefore condemned to having to remember that α-hemolysin/α-toxin causes β-hemolysis. α-toxin is a beta-barrel pore-forming toxin expressed by 95–100% of *S. aureus* isolates from various sites [53,64] and, in our small cohort of strains, 94% of ocular *S. aureus* isolates (Table 1). α-toxin has been widely studied for its role in the pathogenesis of central nervous system infections [65], endocarditis [66], endophthalmitis [52], keratitis [55,67,68], mastitis [69], pneumonia [70,71], sepsis [72,73], and skin and soft tissue infections [74,75]. 

Monomers of α-toxin combine in the eukaryotic cell membrane to form a seven member (heptameric) pre-pore ring-like structure [76]. The mature pore functions as an ion channel, allowing passage of Ca^2+^, K^+^, ATP and other 1–4 kDa-sized molecules into and out of the cell [77,78]. α-toxin targets cells by binding to the receptor a
disintegrin and metalloproteinase 10 (ADAM10), which is required for α-toxin-mediated cytotoxicity [73,76]. Binding of α-toxin to ADAM10 results in assembly of an α-toxin-ADAM10 complex in cholesterol/sphingolipid-rich caveolar rafts [76]. At high concentrations of α-toxin, cell death predominates. However, at sublytic concentrations, α-toxin binding of ADAM10 induces the activation of ADAM10 metalloprotease activity which cleaves E-cadherin adherens junctions and results in disruption of focal adhesions and disruption of tissue barriers [76]. Also, the α-toxin/ADAM10 complex stimulates the dephosphorylation of FAD, p130cas, paxillin and src, leading to the tissue disruption that is characteristic of *S. aureus* infections [5] and enhanced bacterial dissemination [60].

A characteristic feature of *S. aureus* infections of the cornea are epithelial defects (Figure 3). In an experimental rabbit model of *S. aureus* keratitis [67], as organisms replicate and form microcolonies, the corneal epithelial cells die in a zone around the colonies, exposing the underlying stroma, and forming a painful ulcer that is similar to that seen in human cases (Figure 3). The role of α-toxin as a major toxin responsible for corneal epithelial ulceration in keratitis was first demonstrated in an experimental rabbit model in which isogenic mutants of the *S. aureus* strain 8325-4 deficient in α-toxin were injected directly into the corneal stroma. At 15 hours post infection, eyes injected with α-toxin-deficient *S. aureus* had lower slit lamp scores, no epithelial erosions, and less inflammation as indicated by reduced myeloperoxidase activity, compared to the wild-type *S. aureus* parent strain [67]. Further work using this experimental model and an α-toxin-deficient isogenic mutant confirmed that the wild-type strain produced a more severe pathology than the α-toxin mutant [68]. Similar results were observed in the same rabbit model when infections were initiated by wild-type strain Newman and its α-toxin-deficient mutant [79], and in a comparison of *S. aureus* 8325-4 and its α-toxin-deficient mutant in young (6–7 weeks old) and aged mice (36–48 weeks old) using the corneal scarification method [80]. In the latter study, wild-type *S. aureus* produced a more severe pathology than the α-toxin deficient strain, and the pathology was more severe in the aged mice. Callegan et al. [52] used an intravitreal injection model of endophthalmitis in rabbits to demonstrate that infection with an α-toxin-deficient mutant of *S. aureus* 8325-4 resulted in significantly reduced retinal damage compared to the wild-type strain. Recently, Putra et al. [81] reported their comparison of *S. aureus* strain JE2 and its α-toxin-deficient mutant in an experimental mouse model of keratitis initiated after total corneal epithelial debridement. Corneal wound healing was significantly improved following infection with the α-toxin-deficient mutant strain. This mutant was also inefficient in infecting debrided mouse corneas, further confirming the important role of α-toxin in *S. aureus* keratitis. Although it is clear that an absence of α-toxin results in less pathology during *S. aureus* keratitis, it is not known whether α-toxin is acting toward cells or triggering the immune response by similar mechanisms in different species. Different mouse strains have different responses following experimental exposure to *S. aureus*, but whether α-toxin contributes to these differences has not been investigated. 

Topical application of purified α-toxin to the eyes of rabbits caused significant and dose-dependent inflammation of the conjunctiva and iris at six hours post application [68]. Injection of purified α-toxin into rabbit corneas also caused significant and dose-dependent inflammation of the cornea and iris [68]. These results were confirmed in the mouse keratitis model after corneal scarification. Again, corneal pathology was more severe in aged mice (36–38 weeks old) than in young mice (6–7 weeks old) [80]. However, when purified α-toxin was injected into the corneas of young (6–8 weeks old) and aged rabbits (about 30 months old), there was significantly greater disease in young rabbits than in aged ones [84]. Histological studies of rabbit eyes four hours after α-toxin injection showed epithelial cell death by necrosis and apoptosis, sloughing of viable cornea epithelial cells, severe corneal edema, and inflammatory cell migration from the tear film and from the limbal vessels into the cornea [85]. Intravitreal injection of purified α-toxin in a mouse model of endophthalmitis produced an inflammatory response, particularly via IL-1β, but only mild retinal damage and edema, and produced no significant decline in retinal function, as measured by a-or b-wave amplitudes via electroretinography [86].

The expression of α-toxin is tightly regulated. Control of toxin expression in vitro is influenced by the interplay of at least three global regulatory loci: the quorum-sensing systems *agr* (accessory gene regulator), *sarA* (staphylococcal accessory gene regulator), and *sae* (staphylococcal accessory protein effector). Loss of any one of these loci negatively impacts toxin production in vitro [87,88]. During in vivo growth, regulation appears to be more complicated. Booth et al. [89] reported that infection with *S. aureus* strain RN6390 with a defective *agr* resulted in a significant reduction in retinal pathology in a rabbit model of endophthalmitis. Infection with a double mutant of *agr* and *sar* resulted in an even greater reduction of pathology, but a deficiency in *sar* alone resulted in ocular pathology similar to that of the wild-type *S. aureus*. Using an implanted device model in guinea pigs, Goerke et al. [87] reported that mutation of *agr* or *sarA* in strains RN6390 and Newman did not impact the level of transcription of the gene for α-toxin, *hla*. However, loss of *sae* caused an almost total downregulation of α-toxin production in both strains. Xiong et al. [88] examined the α-toxin production of two *S. aureus* strains: RN6390, which lacks a functional *sigB* regulon, and SH1000, which has a repaired *sigB* regulon. *SigB* negatively influences the production of α-toxin in vitro [90]. Xiong et al. [88] reported that in vitro isogenic mutants in RN6390 of *agr*, *sarA*, and *agr/sarA* had significant reductions in α-toxin production. Mutation of *sae* completely abolished α-toxin production in both strains during in vitro growth. SH1000 produced less α-toxin overall than RN6390 but the results of the mutations were similar, except that mutating *sarA* did not reduce α-toxin production. During in vivo growth with RN6390, using an infective endocarditis model in rabbits, only mutation of *sae* resulted in a significant reduction in α-toxin production, compared to the wild-type strain. With SH1000 there was no significant reduction in toxin production with any of the mutant strains. Even with the *sae* mutation, both strains produced measurable amounts of αtoxin during in vivo growth. These data indicate that during in vivo growth other factors, in addition to *agr*, *sarA* and *sae*, are involved in α-toxin regulation. O’Callaghan et al. [68] injected *agr-*deficient mutants of strain 8325-4 into the corneas of rabbits and reported that the *agr* mutant was even less virulent than the α-toxin mutant. Girgis et al. [80] also reported that the *agr* mutant was less virulent than the α-toxin mutant in the mouse keratitis scarification model. Both authors surmised that these outcomes were likely due to deficiencies in α-toxin and all the other toxins under its control in the *agr*-deficient mutants.

The aforementioned studies confirm an important role for α-toxin in the pathogenesis of *S. aureus* keratitis and endophthalmitis. A rational strategy for improving the therapeutic outcome of keratitis would be to block the effect of α-toxin on the cornea. As nearly all ocular *S. aureus* isolates have been reported to possess the *hla* gene (Table 1 and [64]), this virulence factor is a viable target. Passive and active immunization of rabbits against α-toxin has been shown to be effective at reducing corneal disease in the rabbit model of keratitis [79]. Active immunization against α-toxin prevented the formation of corneal epithelial erosions [79]. Whether immunization strategies would be effective in limiting the effects of α-toxin in endophthalmitis or other ocular *S. aureus* infections remains an open question. Recently, nanoparticles composed of erythrocyte membranes surrounding a biologically inert poly-lactic-co-glycolic core have been developed to function as a type of decoy to neutralize pore-forming toxins [91]. These nanoparticles, termed nanosponges, neutralized α-toxin and reduced hemolytic activity in vitro, protected mice from developing staphylococcal α-toxin induced skin lesions, and decreased mortality after systemic injection of a lethal dose of α-toxin [91]. In a mouse model of *S. aureus* endophthalmitis, Coburn et al. [92] reported that intraocular injection of nanosponges alone following infection did not result in improved retinal function compared to untreated mice. However, injection of nanosponges in conjunction with the fourth-generation fluoroquinolone gatifloxacin resulted in decreased inflammation, less damage to the retinal architecture, and significantly improved retinal function compared to untreated or gatifloxacin only treated mice following intraocular infection with an MRSA ocular isolate [92]. These studies suggested that nanosponges might be a viable adjunctive therapy for intraocular infections caused by *S. aureus*. Together, these therapeutics studies imply that blocking α-toxin would improve the therapeutic outcome of ocular *S. aureus* infections.

## 4. Bi-Component Toxins

In *S. aureus*, bi-component leukotoxins are composed of a pair of proteins designated S (slow) and F (fast) based on their elution speeds [93]. Initial binding of the leukotoxin receptor is mediated by the S-component, followed secondarily by binding of the F-component, resulting in formation of a lytic pore-forming octamer (β-barrel) of alternating subunits that insert into the cell membrane, resulting in osmotic imbalance and cell lysis [56,94]. Human *S. aureus* isolates can encode up to five leukocidin toxins: Panton–Valentine leukocidin (PVL), γ-hemolysin AB (HlgAB), γ-hemolysin CB (HlgCB), LukED, and LukAB (also referred to as LukGH) [94]. Elucidation of the role of these toxins in human disease has been hampered by the species and cell specificities of some of the toxins [95].

**Panton–Valentine leukocidin (PVL)** is a prophage-encoded toxin present in about 5% of *S. aureus* isolates and most CA-MRSA strains [53]. PVL binds to complement component C5a anaphylatoxin chemotactic receptors to target neutrophils, monocytes, macrophages, natural killer cells, dendritic cells, and T lymphocytes [60,94]. C5aR1 is highly expressed on phagocytic cells, but C5aR2 has low expression on neutrophils [94,95,96]. The cytotoxicity of PVL can be blocked with C5aR1 antagonists [94,96]. PVL is active against human and rabbit neutrophils, but macaque, cow, and mouse neutrophils are resistant to its effects [93]. PVL stimulates inflammasome activation in monocytes and primary macrophages at sublethal doses [58]. *S. aureus* β-toxin, δ-toxin, γ-toxin, LukDE, and PSMα3 synergize with PVL to amplify IL-1β release to trigger inflammation [58]. Based on species specificity, the most appropriate animal models for the study of PVL are in rabbits and humanized mice with engrafted primary human haematopoietic cells [95].

Not deterred by an acknowledged lack of lytic activity by PVL against mouse neutrophils, Zaidi et al. [97] tested PVL-deficient isogenic mutants of *S. aureus* USA400 and USA300 (currently the predominant CA-MRSA strain in the United States) in an experimental scratch model of mouse keratitis. Inactivation of PVL in the LAC strain of USA300 resulted in reduced corneal opacity and reduced corneal bacterial counts compared to the wild-type strain. However, results with USA400 strains depended on the strain tested. Infection with the isogenic PVL-deficient mutant of MW2 and its wild-type parent strain resulted in similar degrees of corneal opacity and corneal bacterial counts. Infection with the isogenic PVL-deficient mutant of NRS 193 had reduced corneal opacity and reduced corneal bacterial counts compared to its wild-type parent. Infection with the isogenic PVL-deficient mutant of NRS 194 had reduced corneal opacity, but did not have reduced corneal bacterial counts, as compared to its wild-type parent strain. In this study, topical applications of neutralizing polyclonal antibodies against PVL resulted in significantly reduced corneal opacity and bacterial numbers in corneas infected with USA300 strains LAC and SF8300, but in corneas infected with USA400 strains MW2 and NRS 193, the bacterial loads were unaffected and there was no reduction in corneal opacity. Corneas infected with strain NRS 194 showed a reduction in corneal opacity, but bacterial counts were unaffected following the application of anti-PVL antibodies [97].

Siqueira et al. [98] demonstrated that intravitreal injections of six different combinations of the PVL and γ-toxin subunits (S and F) from *S. aureus* ATCC strain 49,775 caused acute inflammatory reactions involving the posterior and anterior chambers as well as the conjunctiva and eyelids. The clinical and histological signs of inflammation began by 4 h after injection and persisted for the 5 days of the experiment. The toxicity was dose-dependent. Certain combinations were more toxic than others. The most virulent combination was the PV-S subunit plus γ-toxin-F subunit, although it is not known if this combination occurs naturally. Liu et al. [99] injected PVL into rabbit eyes and identified retinal ganglion cells as the primary targets of the toxin, and these cells were the only ones identified to express C5a receptors. Binding of PVL triggered increased IL-6 expression in the retina and apoptosis of microglial cells. Liu et al. [100] demonstrated similar results using a rabbit retinal explant model. However, in addition to PVL binding to retinal ganglion cells, the toxin also co-localized to horizontal cells which did not express C5a receptor, and there was no significant increase in IL-6 as compared to the control explants. Peterson et al. [101] recently reported that the PVL gene was detected in only 14.7% of *S. aureus* keratitis isolates in their study, so it is unclear how clinically relevant this toxin is to ocular disease.

γ-toxin consists of two bicomponent pore-forming toxins: HlgAB and HlgCB, which are encoded in the core genome and are reported to be present in 99.8% of clinical isolates [53,102]. During in vitro culture, γ-toxin is highly upregulated in the presence of blood [103] and upon phagocytosis by neutrophils [104]. HlgAB targets the chemokine receptors CXCR1, CXCR2, and CCR2 [105], while HlgCB targets the compliment component C5a receptors C5aR1 and C5aR2 [97]. These receptors are highly expressed on phagocytic cells [105]. HlgCB is active against neutrophils of human, rabbit, macaque, and cow [105]. Mouse neutrophils are resistant to HlgCB. In mice only CCR2 is able to act as a receptor for HlgAB [105]. 

Supersac et al. [106] published a study on the role of γ-toxin using a rabbit model of endophthalmitis infected with the *S. aureus* strain Newman, which the authors believed did not produce α-toxin or β-toxin [107]. After intravitreal injection of either a γ-toxin mutant of Newman, the wild-type, or a γ-toxin complemented strain, all three strains produced a strong inflammatory response, with the only difference being that the γ-toxin mutant did not cause inflammation of the eyelid [106]. More recent studies have shown that while strain Newman has a truncated copy of β-toxin, it does have the α-toxin gene [108], is a weak producer of α-toxin in vitro [79,87], and is a strong producer of α-toxin in vivo [87]. Using the rabbit model of endophthalmitis, Callegan et al. [52] showed with an isogenic mutant of γ-toxin in strain 8325-4 that loss of this gene had no positive effect on retinal function. Dajcs et al. [79], using a rabbit model of keratitis and a γ-toxin mutant of the Newman strain, reported significantly lower slit lamp scores and reduced numbers of PMN in the corneal stroma as compared to the wild-type strain or the genetically rescued strain. Outcomes here appear to be model- and strain-dependent.

Phenol-soluble modulins (PSMs), of which δ-toxin is a member, are a family of small (20–44 amino acids) amphipathic secreted peptides which are only produced by the members of the genus *Staphylococcus* [109]. All *S. aureus* have the *hld* gene and two loci with genes coding for *psm* α and *psmβ* [110]. However, the widely used *S. aureus* strain 8325-4 is unique for not having the gene for *psm*α [111]. PSMs have multiple roles in *S. aureus* pathogenesis, such as facilitating biofilm dissemination, cytolytic activity, and proinflammatory activity [110,112]. PSMα induces the release of TLR2-activating lipoproteins from bacterial cells [113]. While micromolar concentrations of PSMs cause cytolysis, nanomolar concentrations cause inflammation [114]. PSMs are detected by the formyl peptide receptor 2 (FPR2) [114,115] on mammalian cells, resulting in the attraction and activation of neutrophils and inflammation [110]. The mechanism of cytolysis is not clear, but PSMs appear to have a detergent-like action on the cell membrane [112]. Cytolysis is likely nonspecific and receptor independent [116]. Many species of staphylococci express PSM, but *S. aureus* expresses highly cytotoxic forms, such as PSMα1–α4 [116]. Although PSMs are among the most abundant proteins secreted in staphylococcal culture filtrates [109], we are unaware of any studies using animal models to examine their role in eye infections.

## 5. Enzymes

β-toxin is a neutral sphingomyelinase, not a hemolytic toxin per se. β-toxin is able to hydrolyze the plasma membrane lipid sphingomyelin [117] and is responsible for α-hemolysis on blood agar plates. Reports on the frequency of the *hlb* gene in *S. aureus* clinical isolates vary. Although the *hlb* gene was reported to be present in 39–57% of clinical isolates [64,118] and, in our small cohort, 50% of ocular isolates (Table 1), van Wamel et al. [119] reported that in 90% of the isolates examined, the β-toxin gene had been inactivated by phage φSa3 insertion. However, Salgado-Pabon et al. [120] reported that during in vivo and in vitro growth, phage φSa3 can excise from a subpopulation of *S. aureus* and translocate to atypical sites in the chromosome, restoring β-toxin production. β-toxin is toxic to human monocytes, but is inactive at equal concentrations against human erythrocytes, fibroblasts, granulocytes, and lymphocytes [121].

β-toxin has not been shown epidemiologically to be associated with a specific disease or infection severity criteria [112]. However, O’Callaghan et al. [68] demonstrated in a rabbit model of keratitis that topical application of β-toxin caused significant conjunctival inflammation, and corneal injection of purified β-toxin resulted in a rapid edematous reaction in the sclera. Intrastromal injection of an isogenic β-toxin-deficient mutant of *S. aureus* strain 8325-4 caused less scleral edema than its wild-type parental strain, but slit lamp scores, epithelial erosions, and intrastromal ulcers were similar to that of keratitis cause by the wild-type strain [68]. Similar results were reported in an experimental rabbit model of endophthalmitis. *S. aureus* 8325-4 and the same β-toxin-deficient mutant were compared for their virulence following intravitreal injection, and the β-toxin-deficient mutant resulted in significantly reduced retinal damage compared to the wild-type strain [52]. This reduction in damage, however, was not as significant as the reduction of damage in eyes infected with the α-toxin-deficient mutant of the same wild-type parent strain. Together, these studies suggest that β-toxin may contribute to ocular changes in the cornea and within the eye, but its expression in the eye is not essential for complete virulence.

## 6. Discussion

The role of α-toxin in eye infections has been well established by studies comparing α-toxin-deficient isogenic mutants with their wild-type parental strains in models of keratitis and endophthalmitis in both rabbits and mice (Table 2). Ocular administration of purified α-toxin has also been tested. α-toxin’s role as a major virulence factor in ocular *S. aureus* infections is clear cut. However, the studies of the role of the bi-component toxins in ocular infections have been less conclusive. Studies examining the role of PVL were conducted in the mouse [97], the cells of which have been reported to be resistant to PVL [105]. The studies by Supersac et al. [106] and Dajcs et al. [79] both used a γ-toxin-deficient isogenic mutant of the Newman strain in a rabbit model. While Supersac et al. [106] reported a strong inflammatory response following intravitreal injection of wild-type or γ-toxin-deficient Newman, Dajcs et al. [79] reported significantly less inflammation following instrastromal injection of the same γ-toxin-deficient Newman mutant. These two studies suggest a model-dependent role for the virulence of γ-toxin. Studies of the importance of the enzyme β-toxin in corneal and intraocular infection models have reported less virulence in β-toxin-deficient mutants [67]; however, the extent to which ocular virulence is muted in β-toxin-deficient or γ-toxin-deficient ocular infections is not nearly as significant as that of ocular infections caused by α-toxin-deficient *S. aureus.*

Except for α-toxin, our understanding of the role of *S. aureus* toxins in eye infections is limited. Limitations that should be accounted for in such studies are the species-specific sensitivities to several of the bi-component toxins, and the variability in toxin types and the amounts produced by the various isolates commonly used to study *S. aureus* infections. Many of the studies reviewed in this paper used *S. aureus* isolate 8325-4, which has a published genome, numerous mutants, and has been tested in many non-ocular infection models. *S. aureus* 8325-4 was isolated in 1960, before the development of CA-MRSA and HA-MRSA, has been phage cured, is known to lack phenol soluble modulin α3, and has shown significant variation in α-toxin production depending on which lab isolate is used [111]. The use of clinically relevant ocular isolates could provide insight into which genes are important in eye infections and give insight into the interplay of the various toxins produced by the pathogens. However, isolates should first be characterized and compared for relevance to the disease model. Most of the data reviewed in this paper was generated using animal models of infection (Table 2). Although obvious, the many physiological similarities and differences must be accounted for when interpreting and translating data to humans [122,123,124]. All animal infection models have limitations, but most can provide useful information. Animal models not only provide us with the control necessary to initiate various aspects of the infection process, but also facilitate study of the host response to infection, an essential player absent in cell or tissue culture models [125]. Understanding the limitations of these models is important in evaluating the usefulness of the data it produces. For example, rabbit keratitis models using intrastromal injections of bacteria are technically simple, highly controlled, and the output is highly reproducible [55]. However, this model does not replicate the normal manner in which humans develop bacterial keratitis and it bypasses important steps in the infection process. Mouse models of bacterial keratitis suffer from the limitation that only certain mouse strains reliably develop infections [80]. Some mouse strains clear the infections without treatment, unlike infections in humans. The debridement of the mouse corneal epithelium to prepare it for application of bacteria [81] is also not the most accurate model of how the human cornea is damaged prior to infection, unless there is a large wound from chemical or mechanical trauma. To date, the scratch model, similar to that which is used in studies of *Pseudomonas* keratitis [126], is as close an approximation of what may happen to a human cornea prior to *S. aureus* infection that has been attempted. Mouse and rabbit models of bacterial endophthalmitis initiated by intravitreal injections are an approximation of traumatic endophthalmitis, but they lack the traumatic aspect, which may include significant tissue damage and the presence of blood in the eye [125]. Injection or topical application of purified toxin (Table 3) is a strategy that may approximate the toxin concentrations needed for tissue damage and may be helpful in studying the mechanisms by which toxins affect these tissues, but these studies may lack physiological relevance because the amounts and timing of toxin production have not been quantified, bacteria do not produce a single bolus of toxin in infected tissue, and different strains produce different complements of toxin during infection.

A common and relatively straightforward method for examining the importance of a toxin to the pathogenesis of a disease is to create an isogenic mutant specifically deficient in that toxin and examine how the pathology of the disease is altered in the preferred model. While the absence of a single toxin may significantly alter pathogenicity if an organism produces few toxins, for *S. aureus* and many other pathogens, this strategy is rather like removing one part of a car to determine how much damage it does when it hits an object. This practice will provide us with valuable information about how important some parts of the car are, but it will vastly underestimate the importance of most of them. However, an informative start has been made into understanding the roles of *S. aureus* toxins in eye infections. The toxins of *S. aureus* appear to have robust cytotoxic actions on ocular tissues and perhaps subtoxic activities in targeting the both the innate and adaptive branches of the immune system to both modulate the immune response and to block the development of a protective response. A more complete understanding of these activities will be facilitated by developing and using models that are more physiologically relevant and approach the environments and mechanisms by which *S. aureus* infects the human eye.

## Figures and Tables

**Figure 1 toxins-11-00356-f001:**
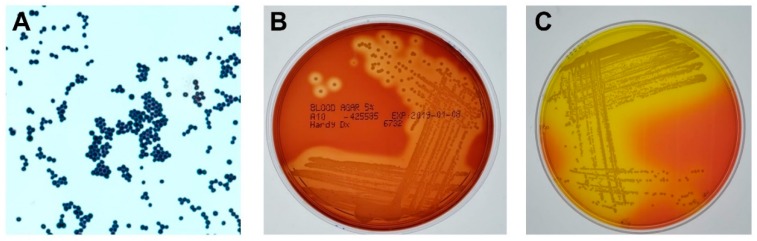
Phenotypes of *Staphylococcus aureus (**S. aureus)*. (**A**) Gram stain of *S. aureus*. Magnification, 100×. (**B**) *S. aureus* grown on 5% sheep blood agar overnight at 37 °C. Note the characteristic zones of β-hemolysis surrounding each colony. (**C**) *S. aureus* grown on mannitol salt agar overnight at 37 °C. *S. aureus*, unlike other staphylococci, ferments mannitol, resulting in acid production and the classic yellow halo within the deep pink agar.

**Figure 2 toxins-11-00356-f002:**
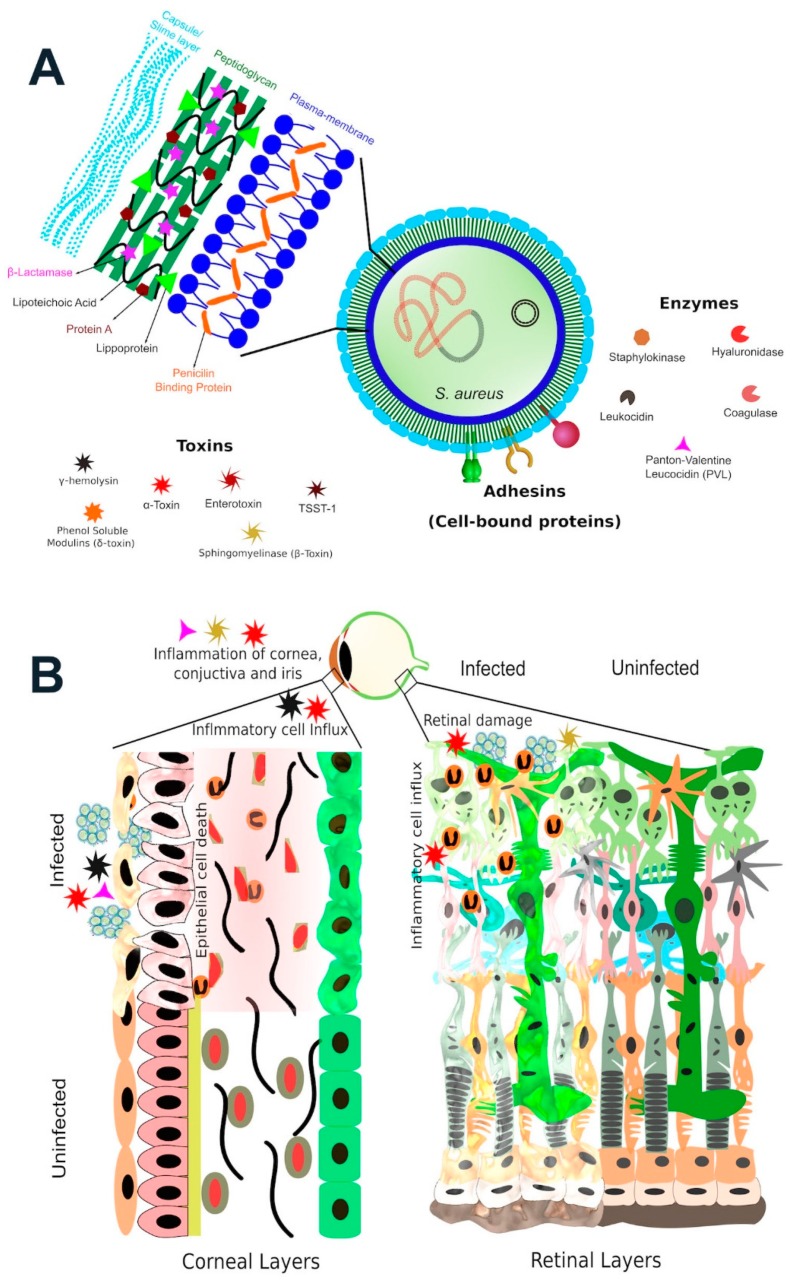
*S. aureus* virulence factors contribute to the pathogenesis of ocular infections. (**A**) *S. aureus* contains numerous virulence factors in its armamentarium which assists in antibiotic resistance, adherence to tissue, interactions with innate immunity, and spread of infection. (**B**) *S. aureus* toxins which have been analyzed in experimental infection models include α-toxin, β-toxin, γ-toxin, and Panton–Valentine leukocidin (PVL). The contribution of these toxins to ocular infections is strain-and model-dependent, as discussed below.

**Figure 3 toxins-11-00356-f003:**
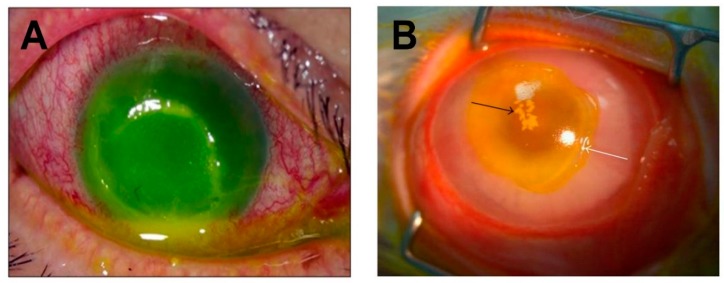
***S. aureus*-induced corneal ulcers in humans and rabbits.** (**A**) A corneal ulcer with superficial stromal infiltration and anterior chamber hypopyon in a human eye infected with MRSA. Copyright © Lee et al., 2010 [82]. (**B**) *S. aureus* keratitis in a rabbit eye. Black arrow indicates inflammatory cell infiltration and staphylococcal microcolonies formed in the corneal stroma. White arrow indicates the edge of the epithelial erosion stained with fluorescein. Copyright © Marquart et al., 2011 [83]. These are open access articles distributed under the Creative Commons Attribution License, which permits unrestricted use, distribution, and reproduction in any medium, provided the original work is properly cited.

**Table 1 toxins-11-00356-t001:** Evaluation of toxin genes in a random sampling of *Staphylococcus aureus* isolated from ocular infections at the Dean A. McGee Eye Institute (USA) from April 2011 to February 2018.

Gene	MSSA (7)	MRSA (9)	All (16)
*hla*	6 (86%)	9 (100%)	15 (94%)
*hlb*	4 (57%)	5 (56%)	8 (50%)
*mecA*	0	9	9

Numbers in parentheses in the top row indicate total number of isolates. Percentages in the table indicate percent positive of the column total by PCR, using primers described previously [64]. PCR for *mecA* was conducted to confirm clinical laboratory results. MSSA, methicillin-sensitive *Staphylococcus aureus*; MRSA, methicillin-resistant *Staphylococcus aureus.*

**Table 2 toxins-11-00356-t002:** Studies analyzing the role of *S. aureus* toxins in ocular infections using staphylococcal strains specifically deficient in the toxin of interest.

Toxin Absent	*S. Aureus* Strain	Ocular Infection Model	Result	References
α-toxin	8325-4	Keratitis rabbit	🡫 Slit lamp scores 🡫 Myeloperoxidase activity 🡫 Corneal erosions	[67]
α-toxin	8325-4	Keratitis rabbit	🡫 Slit lamp scores 🡫 Inflammation	[68]
β-toxin	8325-4	Keratitis rabbit	🡫 Scleral edema	[68]
γ-toxin	Newman	Endophthalmitis rabbit	🡫 Lid inflammation	[106]
α-toxin	Newman	Keratitis rabbit	🡫 Slit lamp scores 🡫 Inflammation	[81]
γ-toxin	Newman	Keratitis rabbit	🡫 Slit lamp scores 🡫 Corneal PMN	[81]
α-toxin	8325-4	Endophthalmitis rabbit	🡫 Retinal damage	[52]
β-toxin	8325-4	Endophthalmitis rabbit	🡫 Retinal damage	[52]
γ-toxin	8325-4	Endophthalmitis rabbit	No change	[52]
α-toxin	8325-4	Keratitis mouse	More severe in aged mice	[80]
PVL	Various USA 300 and 400 strains	Keratitis mouse	Enhanced virulence in a subset of MRSA strains	[97]
α-toxin	JE2	Keratitis mouse	🡡 Corneal healing	[81]

**Table 3 toxins-11-00356-t003:** Summary of studies analyzing the role of *S. aureus* toxins on ocular tissue using purified toxins applied directly to the eye.

Toxin	Inoculation Method	Ocular Model	Result	References
α-toxin	Topical	Rabbit cornea	Inflammation of conjunctiva and iris	[68]
α-toxin	Injection	Rabbit cornea	Inflammation of cornea and iris, corneal epithelial defect	[68]
β-toxin	Topical	Rabbit cornea	Inflammation of conjunctiva	[68]
β-toxin	Injection	Rabbit cornea	Scleral edema	[68]
γ-toxin	Injection	Rabbit cornea	Acute inflammatory reactions	[98]
PVL	Injection	Rabbit cornea	Acute inflammatory reactions	[98]
α-toxin	Injection	Rabbit cornea	↑SLE score, edema, epithelia cell death	[85]
α-toxin	Topical	Mouse cornea	Corneal pathology more severe in aged mice	[80]
α-toxin	Injection	Rabbit cornea	Corneal pathology more severe in young rabbits	[81]
α-toxin	Injection	Mouse Vitreous	Mild retinal damage, no reduction in retinal function	[89]

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
