# Peer review of "An Eye on Staphylococcus aureus Toxins: Roles in Ocular Damage and Inflammation"

_toxins, 2019, doi:10.3390/toxins11060356_

Round 1

Reviewer 1 Report

I enjoyed this very well written article. The subject of ocular S.aureus infection has been carefully analyzed, especially from a pathophysiological perspective which focuses on bacterial toxins, and a complete, critical overview of the literature is provided. I have no changes to suggest to the manuscript in its present form.

Author Response

Reviewer 1: I enjoyed this very well written article. The subject of ocular S. aureus infection has been carefully analyzed, especially from a pathophysiological perspective which focuses on bacterial toxins, and a complete, critical overview of the literature is provided. I have no changes to suggest to the manuscript in its present form.

We thank Reviewer 1 for the positive comments. We made no changes to the manuscript in response to Reviewer 1’s comments.

Reviewer 2 Report

            Thank you for the opportunity to review the review manuscript entitled “Staphylococcus aureus toxins: Roles in ocular damage and inflammation.” The authors in this review manuscript examine experimental and clinical evidence for the pathogenesis of S. aureus eye infections related to the effects of toxins produced by the bacterium.  After a brief historical introduction about S. aureus and development of our understanding of the importance of toxins generally in infection, the authors review the epidemiology of S. aureus eye infections, including complications of eye surgery.  They then include a section on membrane-damaging toxins such as alpha-toxin and various animal models demonstrating its independent toxicity. They describe the regulation of alpha-toxin expression and the potential for alpha-toxin based vaccines.  The authors then describe the potential importance of bi-component toxins, including PVL and gamma-toxin, in eye infections.  The phenol-soluble modulins lack any data on their impact in eye disease. There is a final section on beta-toxin, which appears independently to cause damage when applied topically to the conjunctivae in rabbits. The authors provide 2 clear tables of studies for the use of toxin-gene deficient strains (Table 2) and direct use of purified toxin (Table 3) in models of eye infections.

            Overall, this is a well-written review on a subject for which the literature is rarely summarized.  It also provides a review of the limitations of animal models in the assessment of toxin activity in S. aureus infections. I feel that the authors did a nice job summarizing a sizable literature fairly concisely.

            Because the manuscript is well written and does include most relevant literature, I have the following few, specific comments and questions for the authors, some of them minor:

1. Lines 88-92.  CA-MRSA is composed of many clones around the world.  USA400 was likely common in the US and Canada, but seemed to disappear completely and was replaced by USA300 in North America.  USA300, however, is a relatively rare in most other parts of the world, where unrelated strain types predominate among isolates causing CA-MRSA infections.  Instead, each continent seems to have a predominant type (ST93, ST59, ST80, ST772, and ST30, e.g., depending on the continent). 

2.  Line 96. “scaled”. should be “scalded”

3.  Lines 175-199.  In these mouse studies, do we expect the impact of alpha-toxin to be very different than in human and rabbits?  Can the authors please note that there may be significant differences in mouse models?

4.  Discussion, esp. lines 377-383.  Can the authors consider noting that because many frequently used laboratory strains of S. aureus lack many toxins and that the toxins may interact, future studies are needed with clinically relevant strain types.

5.  Lines 410-415. I like this car analogy!

6.  Introduction (lines 28-98).  This section can be shortened to address more narrowly the relevant background to the manuscript.  

Thank you again for the opportunity to review this interesting manuscript.

Author Response

Reviewer 2: This reviewer summarized and positively commented on the manuscript, but also had concerns that are noted verbatim below.

Because the manuscript is well written and does include most relevant literature, I have the following few, specific comments and questions for the authors, some of them minor:

1. Lines 88-92.  CA-MRSA is composed of many clones around the world.  USA400 was likely common in the US and Canada, but seemed to disappear completely and was replaced by USA300 in North America.  USA300, however, is a relatively rare in most other parts of the world, where unrelated strain types predominate among isolates causing CA-MRSA infections.  Instead, each continent seems to have a predominant type (ST93, ST59, ST80, ST772, and ST30, e.g., depending on the continent). 

We have revised the last paragraph in this section to add this information (page 5 lines 91-94).

2.  Line 96. “scaled”. should be “scalded”

This paragraph was removed from the manuscript.

3.  Lines 175-199.  In these mouse studies, do we expect the impact of alpha-toxin to be very different than in human and rabbits?  Can the authors please note that there may be significant differences in mouse models?

Great questions. Figure 3 certainly suggests that S. aureus keratitis in humans and rabbits is similar with respect to the appearance of corneal ulceration. To answer the question of whether the impact of alpha-toxin is similar in mice, rabbits, and humans, one would have to test these mechanisms in those models. In our opinion, information from mouse models of S. aureus keratitis does not answer these questions. Of course, we cannot mimic these in vivo experiments in humans, but we could test mechanisms in human vs. rabbit vs. mouse primary corneal cultures to begin to answer this question. We addressed this on page 8 lines 182-186 and page 14-15 lines 395-397.

4.  Discussion, esp. lines 377-383.  Can the authors consider noting that because many frequently used laboratory strains of S. aureus lack many toxins and that the toxins may interact, future studies are needed with clinically relevant strain types.

We addressed this comment on pages 13-14, lines 387-395.

5.  Lines 410-415. I like this car analogy!

We do too, thank you!

6.  Introduction (lines 28-98).  This section can be shortened to address more narrowly the relevant background to the manuscript.  

We agree and tried to shorten this section. The summary paragraph was deleted, as well as a sentence at the end of the now last paragraph in that section. We did, however, add sentences in this section based on concern #1 above.

**The reviewers should note that we neglected to include two studies by Liu et al. which analyzed PVL targets in rabbit eyes. This information was added on page 11 lines 295-301.

Toxins EISSN 2072-6651 Published by MDPI AG, Basel, Switzerland RSS E-Mail Table of Contents Alert
Back to Top